# Risk factors associated with *Shigella* diarrhea in 6–35-month-olds: A cross-sectional study, Malawi, 2022–2024

**Vitumbiko P. Yagontha Munthali**[1]*, **Olivia Schultes**[2], **Patricia B. Pavlinac**[2]
**Desiree Witte**[1,3], **Khuzwayo C. Jere**[1,3,4], **Jennifer Cornick**[1,3], **Nigel A. Cunliffe**[3],
**Clement Lefu**[1], **Maureen Ndalama**[1], **Agra Thindwa**[1], **Landwel Mwale**[1], **Latif Ndeketa**[1],
**Stephen Munga**[5], **Donnie Mategula**[1,4,6]

**1** Malawi Liverpool Wellcome Research Programme, Blantyre, Malawi, **2** University of Washington, Seattle, United States of America, **3** Institute of Infection, Veterinary and Ecological Sciences, University of Liverpool, United Kingdom, **4** Kamuzu University of Health Sciences, Blantyre, Malawi. **5** Kenya Medical Research Institute, Kisumu, Kenya, **6** Liverpool School of Tropical Medicine, Liverpool, United Kingdom

* vmunthali@mlw.mw

## Abstract

Diarrhea due to *Shigella* causes 60,000 deaths annually among children under five in low- and middle-income countries. This Enterics for Global Health sub-study examined host, clinical, and environmental factors linked to *Shigella*-attributable diarrhea in Malawian children and assessed seasonal variations. Data from 1,399 children aged 6–35 months presenting with diarrhea at a health centre in Blantyre, Southern Malawi from August 3, 2022, through August 2, 2024, were analyzed. Prevalence ratios of risk factors for *Shigella*-attributable diarrhea were estimated using modified Poisson regression. A *Shigella* case was defined by culture positivity or infection attributable to *Shigella* via quantitative polymerase chain reaction (qPCR). Multivariate analyses adjusted for age, wealth index quintile, and vaccination status. Effect modification by season (rainy vs. dry) was assessed for each predictor. Sensitivity analyses compared cases identified by culture or qPCR. *Shigella*-attributable diarrhea was found in 12.4% of children. Older age (12–35 months), dysentery, and diarrhea severity were positively associated with *Shigella*-attributed diarrhea in unadjusted and adjusted models; stunting and age-appropriate vaccination were significant only in unadjusted models. Seasonal effect modification occurred for wasting and *Shigella* infection in the dry season but not rainy season. Longer diarrhea duration was marginally linked to increased *Shigella* risk in the dry season only. Conversely, open defaecation/ unimproved sanitation increased *Shigella* risk during the rainy season but not the dry season. Sensitivity analyses showed consistent risk patterns for culture- and qPCR-diagnosed cases. This study identified age, dysentery, and diarrhea severity as key factors for Shigella-attributed diarrhea. Seasonal effects influenced the relationships between *Shigella* infection and wasting, diarrhea duration, and sanitation.

**Data availability statement:** De-identified datasets are publicly available on Vivli https://doi.org/10.25934/PR00011860.

**Funding:** This work was supported by the Gates Foundation (INV-031791, INV-045988, INV-062665, and INV-076498 to Patricia B Pavlinac), and the funders had no role in study design, data collection, analysis, decision to publish, or preparation of the manuscript.

**Competing interests:** The authors have declared that no competing interests exist.

Policies prioritizing vulnerable populations and accounting for seasonal variations can help reduce the *Shigella* burden in children.

## Introduction

*Shigella* infection is a significant global health problem, particularly affecting children under five years in low- and middle-income countries [1,2]. *Shigella* is transmitted through the fecal-oral pathway, is highly infectious, and can lead to severe gastro-enteritis, dysentery, and hospitalization, resulting in economic burden for families and society [3–6]. In addition to acute morbidity, *Shigella* infection and other enteric pathogens have been associated with linear growth faltering and stunting in early childhood, as observed in large multicountry birth cohort and case–control studies [7,8]. Early childhood stunting is linked to long-term adverse outcomes, including impaired cognitive development, lower educational attainment, and diminished capacity to reach full developmental potential in adulthood [7,8]. Factors contributing to the high incidence [9] of *Shigella*-attributable diarrhea include malnutrition, poor sanitation, and crowded environments [1,2,10]. Young children are especially vulnerable due to their behaviors, such as touching contaminated surfaces and putting their fingers in their mouths [1,2]. Despite the recognized risks, many aspects of *Shigella* infection remain poorly understood. In particular, the associations between host, clinical, and environmental factors and *Shigella*-attributable diarrhea are not well defined. A better understanding of these risk factors for *Shigella* infection will support the roll out of *Shigella* vaccines currently in development [10].

Although studies have examined seasonal patterns of shigellosis, the relationship between seasonality and factors associated with *Shigella* incidence is not well understood [11–14], highlighting the need for more localized research [15]. Specifically, it is unclear how seasonal variation affects *Shigella* transmission among different populations in Malawi [11,13]. The rainy season often brings flooding, which can contaminate water sources with fecal matter, creating conducive conditions for *Shigella* spread [13]. However, water shortages in the dry season may lead communities to rely on limited water resources which are more prone to contamination, and can increase the risk of infection [13]. Additionally, rising temperatures associated with climatic change may increase the survival and transmission of *Shigella* bacteria in the environment [13]. These complex interactions highlight the need for a deeper understanding of how risk factors influence *Shigella* dynamics, as the precise effects of such factors on transmission and infection rates remain inadequately documented [15].

We explored host, clinical, and environmental factors associated with *Shigella*-attributable diarrhea among children aged 6–35 months in Malawi and assessed whether seasonal variation modified these relationships. We included diagnosis by both culture and quantitative polymerase chain reaction (qPCR) in the risk factor analysis because both culture and PCR are valuable diagnostic approaches for *Shigella* and to show consistent associations between most predictors and *Shigella* across both methods [16].

## Materials and methods

### Ethical consideration

This work is a sub-study of the Enterics for Global Health project. Ethical approval of the study entitled P.10/21/3437 was obtained from the College of Medicine Research Ethics Committee (COMREC). Written informed consent was obtained from the legally accepted representatives (parents or caregivers) of all study participants prior to their enrollment.

### Study design and setting

The Enterics for Global Health (EFGH) *Shigella* surveillance study utilized both a cross-sectional census and longitudinal follow-up of diarrhea cases to estimate *Shigella* incidence and document the impact of *Shigella*-attributable diarrhea across seven countries in Africa, Asia, and Latin America [17]. Leveraging data from this hybrid surveillance initiative, we performed a secondary analysis to explore host, clinical and environmental factors associated with *Shigella* infection. This cross-sectional study focused on identifying correlates of *Shigella* diarrhea among children aged 6–35 months who sought medical care for diarrhea at the EFGH recruitment site in Blantyre, Malawi.

Participants were recruited at the Ndirande Health facility, a key provider of healthcare services in the high-density neighbourhood of Ndirande in Blantyre, Malawi. Children who presented with diarrhea from August 3, 2022, through August 2, 2024 were screened for enrollment [17]. Children aged 6–35 months were eligible if they presented to the health facility and met all of the following criteria: three or more abnormally loose or watery stools in the previous 24 hours, diarrhea had started within the last seven days, the child had been present in the facility for less than 4 hours at the time of screening and the caregiver provided informed written consent for the child's participation in the study [17].

### Data collection

At enrollment, staff conducted a clinical examination, measured height, weight, and mid-upper arm circumference, recorded vaccination data from the child's health passport book (if available), and administered a standard questionnaire to caregivers to capture health history, maternal characteristics and household details [17]. Study staff collected rectal swabs which were tested for *Shigella* using both culture and qPCR. Caregivers were given a diarrhea diary to record symptoms each day for fourteen days following enrollment. The diarrhea diary was returned to the study staff at either the four-week or three-month follow-up visit, and during these visits additional information was sought on symptoms during the index diarrhea episode [17].

### Outcome variable

All participants were tested for *Shigella* using both culture and qPCR methods. The primary outcome was *Shigella* detected by either culture or qPCR, and that in sensitivity analysis, we examined two versions of the outcome, defined as *Shigella* identified by culture and *Shigella* identified by qPCR, respectively [17–19].

### Predictor variables

**Socioeconomic Status.** A household wealth index quintile was calculated using a validated abbreviated version of the Demographic and Health Survey, which is meant to approximate the national distribution of household wealth [20].

### Vaccination status

Age-appropriate vaccination was defined as children who had received all scheduled doses of eight key vaccines (BCG, Polio, DPT, Hepatitis B, *Haemophilus influenzae* type b, Rotavirus, MMR, and pneumococcal pneumonia) according to the national vaccine schedule in Malawi [17], allowing for a one-month delay in receipt of doses.

### Nutritional status

Wasting, stunting, and underweight were calculated according to World Health Organization guidance using weight-for-height z-score and mid-upper arm circumference, height-for-age z-score, and weight-for-age z-score, respectively [21]. Z-scores less than -6 or greater than 6 were considered implausible and were excluded from the analysis.

### Diarrhea severity

Diarrhea episode severity was assessed through direct clinical observation, caregiver-reported symptoms, and the returned diarrhea diary, along with information on vomiting, dysentery, and episode duration. Additionally, two classification measures of diarrhea severity were used: the modified Vesikari score (MVS), with or without dysentery [22], and the Global Enterics Multicenter Study moderate-to-severe diarrhea (GEMS MSD) [8].

### Water, sanitation, and hygiene (WASH)

Household water and sanitation were defined according to the Joint Monitoring Program definitions as improved or unimproved [23], while household hygiene was defined as handwashing with soap.

### Seasonality

When examining seasonal effect modification, participants were assigned to either the rainy (November 1 through April 30) or dry (May 1 through October 31) season based on their date of enrolment [12].

### Statistical analysis

We used Poisson regression to calculate prevalence ratios (PRs) with robust standard errors. For cross-sectional data with common outcomes, using logistic regression can lead to inflated estimates of risk, making modified Poisson regression a more suitable analytic approach [24].

Generalized estimating equations (GEE) were used to account for clustering of children who were enrolled multiple times. Participants with missing predictor or covariate information were dropped from the relevant model.

Univariate analyses were adjusted for enrollment site, while multivariate analyses were adjusted for enrollment site, child age, wealth index quintile, and age-appropriate vaccination status. We tested all predictors for seasonal effect modification and presented stratified results by season. Additionally, a sensitivity analysis was conducted to compare the correlates of *Shigella*-attributable diarrhea identified through culture methods versus qPCR. In both the models with the effect modifier term and in sensitivity analysis, simplified versions of the wealth index and wasting variables were used due to small cell sizes. All analyses were conducted using R version 4.3.

## Results

### Demographic characteristic analysis

Of the 1,399 enrolled children, 173 (12.4%) had *Shigella*-attributable diarrhea. A total of 99 children (7.1%) tested positive for *Shigella* via culture, 131 (9.4%) were identified as *Shigella*-positive by qPCR, and 57 (4.1%) tested positive using both methods. Enrollment was almost evenly split across seasons, with 706 children enrolled during the dry season and 693 during the rainy season (Table 1). Females comprised 47% of participants. The age distribution comprised, 34% aged 6–11 months, 45% aged 12–23 months, and 21% aged 24–35 months. More than half of mothers (56%) had at least some secondary education. More than a third (38.7%) of children had not received age-appropriate vaccinations, while a substantial proportion (31.8%, n = 445) were missing vaccination data. Nutritional assessments showed that 4.6% of children were moderately or severely wasted, 29.0% were stunted, and 12.5% were moderately or severely underweight. During the index diarrhea episode, 10% of children experienced dysentery and 46% had vomiting. Nearly all households

**Table 1. Participant characteristics, Enterics for Global Health sub-study, Malawi 2022-2024.**

| | | Participants seen during rainy season N = 693[1] | Participants seen during dry season N = 706[1] | Overall N = 1,399[1] |
|---|---|---|---|---|
| Age (months) | 6-11 | 233 (34.0) | 245 (35.0) | 478 (34.0) |
| | 12-23 | 309 (45.0) | 318 (45.0) | 627 (45.0) |
| | 24-35 | 151 (22.0) | 143 (20.0) | 294 (21.0) |
| Sex | Female | 305 (44.0) | 346 (49.0) | 651 (47.0) |
| | Male | 388 (56.0) | 360 (51.0) | 748 (53.0) |
| Maternal education | ≤ Primary school | 303 (44.0) | 317 (45.0) | 620 (4.0) |
| | ≥ Secondary | 390 (56.0) | 389 (55.0) | 779 (56.0) |
| Wealth index quintile | 1st (least wealthy) | 78 (11.0) | 80 (11.0) | 158 (11.0) |
| | 2nd | 225 (32.0) | 224 (32.0) | 449 (32.0) |
| | 3rd | 279 (40.0) | 259 (37.0) | 538 (38.0) |
| | 4th | 88 (13.0) | 108 (15.0) | 196 (14.0) |
| | 5th (most wealthy) | 23 (3.3) | 35 (5.0) | 58 (4.1) |
| Age-appropriate vaccinations given | No | 257 (37.1) | 283 (40.1) | 540 (38.6) |
| | Yes | 205 (29.6) | 209 (29.6) | 414 (29.6) |
| | Unknown | 231 (33.3) | 214 (30.3) | 445 (31.8) |
| Wasting | None | 655 (94.6) | 672 (95.2) | 1,327 (94.8) |
| | Moderate | 29 (4.2) | 24 (3.4) | 53 (3.8) |
| | Severe | 4 (0.6) | 7 (1.0) | 11 (0.8) |
| | Unknown | 5 (0.7) | 3 (0.4) | 8 (0.6) |
| Stunting | Not Stunted | 491 (70.9) | 497 (70.4) | 988 (70.6) |
| | Stunted | 197 (28.4) | 205 (29.0) | 402 (28.7) |
| | Unknown | 5 (0.7) | 4 (0.6) | 9 (0.6) |
| Underweight | Not underweight | 605 (87.0) | 619 (88.0) | 1,224 (87.0) |
| | Moderate | 67 (9.7) | 72 (10.0) | 139 (9.9) |
| | Severe | 21 (3.0) | 15 (2.1) | 36 (2.6) |
| Dysentery | No | 616 (89.0) | 645 (91.0) | 1,261 (90.0) |
| | Yes | 77 (11.0) | 61 (8.6) | 138 (9.9) |
| Vomiting | No | 376 (54.0) | 386 (55.0) | 762 (54.0) |
| | Yes | 317 (46.0) | 320 (45.0) | 637 (46.0) |
| Diarrhea duration in days | (Median, IQR) | 3 (3, 5) | 3 (3, 5) | 3 (3, 5) |
| GEMS-MSD[2] | Less-severe | 590 (85.0) | 624 (88.0) | 1,214 (87.0) |
| | Moderate-to-severe | 103 (15.0) | 82 (12.0) | 185 (13.0) |
| MVS +/- dysentery[3] | Mild with no dysentery | 415 (60.0) | 446 (63.0) | 861 (62.0) |
| | Moderate/Severe | 278 (40.0) | 260 (37.0) | 538 (38.0) |
| Household drinking water source | Improved | 691 (100.0) | 701 (99.0) | 1,392 (99.0) |
| | Unimproved/Surface water | 2 (0.3) | 5 (0.7) | 7 (0.5) |
| Household sanitation | Improved | 520 (75.0) | 537 (76.0) | 1,057 (76.0) |
| | Unimproved | 173 (25.0) | 169 (24.0) | 342 (24.0) |
| Household hygiene | Wash hands without soap | 196 (28.3) | 220 (31.2) | 416 (29.7) |
| | Wash hands with soap | 497 (71.7) | 484 (68.6) | 981 (70.1) |
| | Unknown | 0 (0.0) | 2 (0.3) | 2 (0.1) |

[1] n (%); Median (Q1, Q3)

[2] Defined as in Kotloff, Lancet GH, 2019. Moderate-to-severe diarrhea (MSD) defined as presenting to a health facility with diarrhea and severe or some dehydration (by WHO criteria), visible blood in stool, or inpatient admission.

[3] Defined as in Pavlinac, Vaccines, 2022 as a modified Vesikari score (MVS) of 9+ or presence of visible blood in stool.

(99%) reported access to improved drinking water, though 24% used unimproved sanitation facilities or practiced open defecation, and 30% reported handwashing without soap. By household wealth quintiles: 11% of children lived in the first (poorest), 32% in the second, 38% in the third, 14% in the fourth, and 4.1% in the fifth (wealthiest) quintile (Table 1). Most children were only enrolled in EFGH once (n = 1,306); 45 were enrolled twice, and 1 was enrolled three times.

### Univariate and multivariate analyses

Older children were more likely to have *Shigella* than younger children (Fig 1). In univariate analysis, when compared to children aged 6–11 months, children aged 12–23 months were 2.71 times (95% confidence interval [CI]: 1.76, 4.16) more likely to have *Shigella*-attributed diarrhea, and children aged 24–35 months were 3.90 times as likely (95% CI: 2.51, 6.08). Similar results were observed in multivariate models, where children aged 12–23 months had a PR of 2.45 (95% CI: 1.58, 3.79) and children aged 24–35 months had a PR of 3.38 (95% CI: 2.10, 5.44) compared to children aged 6–11 months. Stunting was associated with higher *Shigella* detection in the univariate model (PR 1.35, 95% CI: 1.01, 1.81) but not in the multivariate model (PR 1.03, 95% CI: 0.77, 1.39). Wasting was not associated with *Shigella* in the univariate model but was associated with an increased prevalence of *Shigella* in the multivariate model (moderate wasting PR 1.84, 95% CI 1.04, 3.25 and severe wasting PR 2.18, 95% CI 0.68, 6.98).

Dysentery was strongly associated with increased prevalence of *Shigella* infection in both univariate (PR 1.84, 95% CI: 1.28, 2.65) and multivariate analysis (PR 1.66, 95% CI: 1.16, 2.39), as was moderate-to-severe diarrhea as classified by GEMS-MSD (univariate PR 1.66, 95% CI: 1.18, 2.34; multivariate PR 1.51, 95% CI: 1.08, 2.21). Children who did not

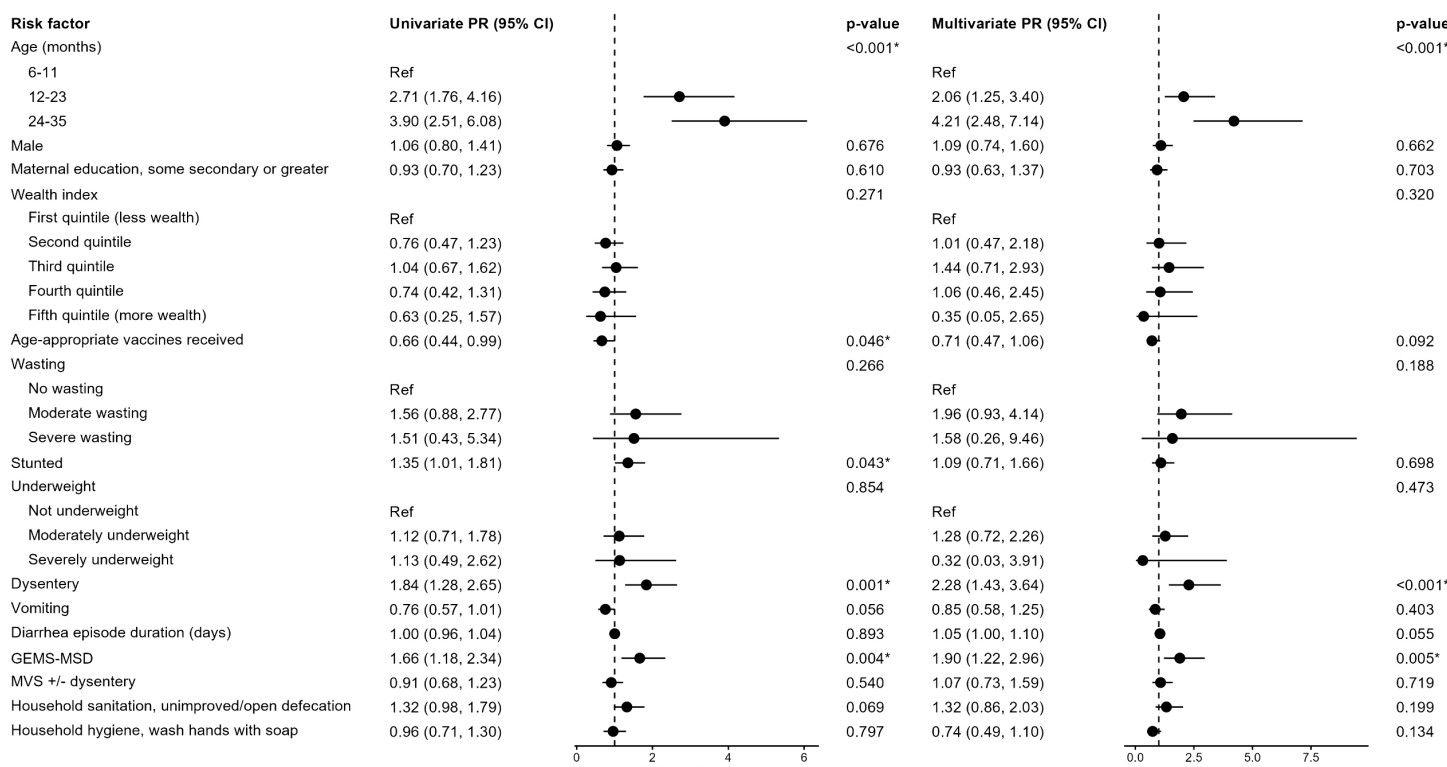

**Fig 1. Prevalence ratios of host, clinical and environmental correlates on the outcome of Shigella ascertained by culture or qPCR.** Multivariate models adjusted for age, wealth index quintile, and age-appropriate vaccination.

receive age-appropriate vaccines and whose vaccination status was unknown were more likely to have *Shigella* compared to those who had received vaccines in the univariate model (PR 1.50, 95% CI: 1.00, 2.26 and PR 2.25, 95% CI 1.52, 3.32, respectively); the association was not significant in the multivariate model (PR 1.39, 95% CI: 0.92, 2.09 and PR 1.64, 95% CI 1.08, 2.49 respectively). We were unable to run models for household drinking water due to the high proportion of participants who reported having access to improved drinking water (99%).

The sensitivity analyses, which disaggregated *Shigella* detected by culture versus qPCR, identified some associations that differed from the combined findings. There was a statistically significant negative association between culture-positive *Shigella* and receipt of all age-appropriate vaccinations (PR 0.49, 95% CI: 0.27, 0.87) (Table 2), similar to the association observed using the primary outcome. In contrast, no significant association was found between receipt of age-appropriate vaccinations and *Shigella* detected by qPCR (PR 0.67, 95% CI: 0.41, 1.08). In the main model, household sanitation was marginally associated with *Shigella* infection (PR 1.32, 95% CI: 0.98, 1.79) (Fig 1). When disaggregated by *Shigella* detection method, there was a significant association between unimproved sanitation and increased culture-positive *Shigella* (PR 1.55, 95% CI: 1.04, 2.31), but no significant association between sanitation and *Shigella* detected by qPCR (PR 1.07, 95% CI: 0.74, 1.57).

Significant seasonal effect modification was observed between *Shigella* and wasting (p-value = 0.033), diarrhea duration (p-value = 0.044), and household sanitation (p-value = 0.046) (Table 3). Wasting was significantly associated with increased *Shigella*-attributable diarrhea in the dry season (PR 2.74, 95% CI 1.45, 5.18) but not in the rainy season (PR 0.81, 95% CI 0.32, 2.05). Longer diarrhea duration was marginally associated with an increased risk of *Shigella* in the dry season (PR 1.07, 95% CI: 1.00, 1.14), but there was no association in the rainy season (PR 0.97, 95% CI: 0.92, 1.03). Unimproved sanitation was associated with an increased risk of *Shigella* in the rainy season (PR 1.68, 95% CI: 1.18, 2.41) but not in the dry season (PR 0.82, 95% CI:.46, 1.46).

## Discussion

This EFGH sub-study provides insights into the burden and risk factors associated with *Shigella*-attributable diarrhea in young children attending a health centre in Malawi. A total of 12.4% of children aged 6–35 months had *Shigella*-attributable diarrhea; *Shigella* was identified in 7.1% of children via culture and in 9.4% via qPCR. The highest burden was observed in children aged 12–24 months, and dysentery, moderate to severe diarrhea and wasting were associated with the presence of *Shigella*. Moderate or severe wasting and diarrhea duration were associated with *Shigella* infection during the dry season, while unimproved sanitation increased risk in the rainy season. Moreover, children with age-appropriate vaccinations were less likely to have *Shigella*-attributable diarrhea, and associations between predictors and *Shigella* infection remained largely consistent across culture- and qPCR-based diagnoses.

Children aged 12–35 months were significantly more likely to be infected with *Shigella* than younger children. This finding aligns with previous literature, including studies by Saha et al. [3] and research from Bangladesh [4], which have shown age-related increases in *Shigella* prevalence and severity. The heightened vulnerability in this age group is likely due to a combination of waning maternal immunity and increased exposure to contaminated food and water as children begin to explore their environment [3,4]. Although this study demonstrated that the highest burden was among older children, *Shigella* was detected in 14.5% of infants aged 6–11 months in this study and 11% of infants aged 6–12 months in other studies [25], emphasizing that all children under two years remain at risk. These findings reinforce the need for early interventions that include vaccination and WASH improvements during infancy and toddlerhood [25].

Dysentery was strongly associated with *Shigella* infection, consistent with findings from Ethiopia and other sub-Saharan regions [5,15]. While this supports clinical reliance on dysentery for presumptive treatment, at least half of *Shigella* cases occur without dysentery in this study. Therefore, focusing exclusively on dysentery could result in underdiagnosis and undertreatment, and future work should explore the development of clinical diagnostic algorithms or point-of-care testing.

**Table 2. Host, clinical and environmental correlates of *Shigella* detected by culture versus qPCR among Malawian children ages 6-35 months.**

| | | *Shigella* + by Culture | | *Shigella* + by qPCR | |
|---|---|---|---|---|---|
| | | Univariate PR2 (95%CI) | Multivariate1 PR2 (95%CI) | Univariate PR2 (95%CI) | Multivariate1 PR2 (95%CI) |
| Age (months) | 6-11 | Ref. | Ref. | Ref. | Ref. |
| | 12-23 | 2.93 (1.57-5.49)*** | 2.56 (1.19-5.53)*** | 3.43 (2.00-5.86)*** | 2.44 (1.36-4.38)*** |
| | 24-35 | 5.59 (2.99-10.50)*** | 7.31 (3.38-15.8)*** | 4.58 (2.61-8.02)*** | 4.44 (2.17-9.06)*** |
| Sex | Female | Ref. | Ref. | Ref. | Ref. |
| | Male | 0.96 (0.66-1.40) | 0.84 (0.51-1.39) | 1.09 (0.78-1.52) | 1.03 (0.64-1.68) |
| Maternal education | ≤ Primary | Ref. | Ref. | Ref. | Ref. |
| | ≥ Secondary | 0.88 (0.60-1.29) | 0.85 (0.51-1.40) | 0.90 (0.65-1.25) | 1.00 (0.63-1.58) |
| Wealth quintile | 1st and 2nd (Less wealthy) | Ref. | Ref. | Ref. | — |
| | 3rd | 1.26 (0.84-1.90) | 1.24 (0.72-2.13) | 1.20 (0.84-1.72) | 1.53 (0.94-2.49) |
| | 4th/5th (More wealthy) | 0.83 (0.46-1.50) | 0.78 (0.35-1.75) | 0.85 (0.52-1.40) | 1.12 (0.57-2.19) |
| Age-appropriate vaccination | No | Ref. | Ref. | Ref. | Ref. |
| | Yes | 0.49 (0.27-0.87)* | 0.54 (0.30-0.97)* | 0.67 (0.41-1.08) | 0.72 (0.44-1.17) |
| | Unknown | 2,72 (1.54-4.80) | 1.81 (0,97-3.38) | 2.19 (1.40-3.43) | 1.55 (0.97-2.49) |
| Wasting | None | Ref. | Ref. | Ref. | Ref. |
| | Moderate | 1.10 (0.43-2.82) | 1.26 (0.31-5.06) | 1.84 (1.00-3.41) | 2.21 (0.97-5.01) |
| | Severe | 1.30 (0.20-8.49) | 2.92 (0.50-17.0) | 1.98 (0.56-7.03) | 2.15 (0.36-12.90) |
| Stunting | Not Stunted | Ref. | Ref. | Ref. | Ref. |
| | Stunted | 1.70 (1.16-2.49)** | 1.37 (0.80-2.36) | 1.47 (1.04-2.06)* | 1.12 (0.69-1.82) |
| Underweight | No | Ref. | Ref. | Ref. | Ref. |
| | Moderate | 1.04 (0.56-1.95) | 1.40 (0.67-2.92) | 1.20 (0.71-2.03) | 1.11 (0.55-2.23) |
| | Severe | 1.59 (0.61-4.12) | 0.71 (0.10-5.06) | 1.17 (0.44-3.14) | 0.24 (0.00-28.2) |
| Dysentery | No | Ref. | Ref. | Ref. | Ref. |
| | Yes | 2.31 (1.47-3.64)*** | 2.19 (1.14-4.20)* | 2.26 (1.52-3/38)*** | 3.04 (1.76-5.27)*** |
| Vomiting | No | Ref. | Ref. | Ref. | Ref. |
| | Yes | 0.65 (0.44-0.97)* | 0.78 (0.46-1.31) | 0.71 (0.50-1.00)* | 0.83 (0.52-1.33) |
| Diarrhea duration | (in days) | 0.98 (0.92-1.05) | 1.04 (0.96-1.12) | 1.03 (0.98-1.07) | 1.08 (1.03-1.13)** |
| 3 GEMS-MSD | Less-severe | Ref. | Ref. | Ref. | Ref. |
| | Moderate-to-severe | 1.99 (1.28-3.08)** | 1.69 (0.91-3.14) | 2.02 (1.38-2.95)*** | 2.60 (1.54-4.39)*** |
| 4MVS +/- dysentery | Mild with no dysentery | Ref. | Ref. | Ref. | Ref. |
| | Moderate/Severe | 0.96 (0.65-1.42) | 0.92 (0.54-1.57) | 1.03 (0.73-1.46) | 1.22 (0.74-2.02) |
| Household sanitation | Improved | Ref. | Ref. | Ref. | Ref. |
| | Unimproved | 1.55 (1.04-2.31)* | 1.67 (0.97-2.87) | 1.07 (0.74-1.57) | 1.05 (0.57-1.93) |
| Household washes hands with soap | No | Ref. | Ref. | Ref. | Ref. |
| | Yes | 0.98 (0.65-1.48) | 0.63 (0.37-1.06) | 1.01 (0.70-1.45) | 0.91 (0.53-1.57) |

[1]Multivariate models adjusted for age, wealth index quintile, and age-appropriate vaccination.

[2]Prevalence ratio with 95% confidence intervals.

* $p \leq 0.05$.

** $p \leq 0.01$.

*** $p \leq 0.001$.

[3]Defined as in Kotloff, Lancet GH, 2019. Moderate-to-severe diarrhea (MSD) defined as presenting to a health facility with diarrhea and severe or some dehydration (by WHO criteria), visible blood in stool, or inpatient admission.

[4]Defined as in Pavlinac, Vaccines, 2022 as a MVS of 9+ or presence of visible blood in stool.

**Table 3. Host, clinical and environmental correlates of *Shigella* ascertained by culture or qPCR, stratified by dry and rainy season.**

| Characteristic | Dry season | | | | | Rainy season | | | | | |
|---|---|---|---|---|---|---|---|---|---|---|---|
| | *Shigella* +1 N=70 | *Shigella* -1 N=636 | PR[2] | 95% CI[2] | p-value | *Shigella* +1 N=103 | *Shigella* -1 N=590 | PR[2] | 95% CI[2] | p-value | Effect Modifier p-value |
| Age (months) | | | | | 0.019 | | | | | 0.000 | 0.709 |
| 6-11 | 11 (16%) | 234 (37%) | — | — | | 14 (14%) | 219 (37%) | — | — | | |
| 12-23 | 38 (54%) | 280 (44%) | 2.60 | 1.14, 5.92 | | 50 (49%) | 259 (44%) | 2.66 | 1.51, 4.68 | | |
| 24-35 | 21 (30%) | 122 (19%) | 3.60 | 1.45, 8.96 | | 39 (38%) | 112 (19%) | 4.29 | 2.43, 7.58 | | |
| Sex | | | | | 0.491 | | | | | 0.310 | 0.267 |
| Female | 37 (53%) | 309 (49%) | — | — | | 41 (40%) | 264 (45%) | — | — | | |
| Male | 33 (47%) | 327 (51%) | 0.85 | 0.54, 1.34 | | 62 (60%) | 326 (55%) | 1.21 | 0.84, 1.73 | | |
| Maternal education | | | | | 0.718 | | | | | 0.579 | 0.927 |
| Primary school or less | 33 (47%) | 284 (45%) | — | — | | 47 (46%) | 256 (43%) | — | — | | |
| Some secondary or greater | 37 (53%) | 352 (55%) | 0.92 | 0.59, 1.44 | | 56 (54%) | 334 (57%) | 0.90 | 0.63, 1.29 | | |
| Wealth index quintile | | | | | 0.000 | | | | | 0.544 | 0.313 |
| First/second quintile (less wealth) | 24 (34%) | 280 (44%) | — | — | | 46 (45%) | 257 (44%) | — | — | | |
| Third quintile | 34 (49%) | 225 (35%) | 1.79 | 1.06, 3.02 | | 44 (43%) | 235 (40%) | 1.03 | 0.71, 1.50 | | |
| Fourth/fifth quintile (more wealth) | 12 (17%) | 131 (21%) | 357 | 229, 556 | | 13 (13%) | 98 (17%) | 0.75 | 0.42, 1.33 | | |
| Age-appropriate vaccination | | | | | 0.023 | | | | | 0.007 | 0.514 |
| Received | 14 (20%) | 195 (31%) | — | — | | 18 (17%) | 185 (31%) | — | — | | |
| Has not received | 25 (36%) | 258 (41%) | 1.32 | 0.70, 2.49 | | 39 (38%) | 220 (37%) | 1.71 | 1.01, 2.90 | | |
| Unknown | 31 (44%) | 183 (29%) | 2.16 | 1.18, 3.95 | | 46 (45%) | 185 (31%) | 2.26 | 1.35, 3.77 | | |
| Wasting | | | | | 0.002 | | | | | 0.650 | 0.033 |
| None | 62 (89%) | 610 (96%) | — | — | | 98 (96%) | 557 (95%) | — | — | | |
| Moderate/Severe | 8 (11%) | 23 (3.6%) | 2.74 | 1.45, 5.18 | | 4 (3.9%) | 29 (4.9%) | 0.81 | 0.32, 2.05 | | |
| Stunting | | | | | 0.078 | | | | | 0.221 | 0.486 |
| Not Stunted | 43 (61%) | 454 (72%) | — | — | | 68 (67%) | 423 (72%) | — | — | | |
| Stunted | 27 (39%) | 178 (28%) | 1.51 | 0.95, 2.40 | | 34 (33%) | 163 (28%) | 1.26 | 0.87, 1.83 | | |
| Underweight | | | | | 0.194 | | | | | 0.827 | 0.186 |
| None | 57 (81%) | 562 (88%) | — | — | | 92 (89%) | 513 (87%) | — | — | | |
| Moderate | 10 (14%) | 62 (9.7%) | 1.49 | 0.77, 2.87 | | 9 (8.7%) | 58 (9.8%) | 0.91 | 0.49, 1.68 | | |
| Severe | 3 (4.3%) | 12 (1.9%) | 2.19 | 0.77, 6.22 | | 2 (1.9%) | 19 (3.2%) | 0.71 | 0.22, 2.34 | | |
| Dysentery | | | | | 0.030 | | | | | 0.006 | 0.974 |
| No | 60 (86%) | 585 (92%) | — | — | | 84 (82%) | 532 (90%) | — | — | | |
| Yes | 10 (14%) | 51 (8.0%) | 2.03 | 1.07, 3.83 | | 19 (18%) | 58 (9.8%) | 1.83 | 1.19, 2.80 | | |
| Vomiting | | | | | 0.089 | | | | | 0.248 | 0.497 |
| No | 45 (64%) | 341 (54%) | — | — | | 61 (59%) | 315 (53%) | — | — | | |
| Yes | 25 (36%) | 295 (46%) | 0.66 | 0.41, 1.06 | | 42 (41%) | 275 (47%) | 0.81 | 0.56, 1.16 | | |
| Diarrhea episode duration (days) | 4 (3, 5) | 3 (3, 5) | 1.07 | 1.00, 1.14 | 0.063 | 3 (2, 5) | 4 (3, 5) | 0.97 | 0.92, 1.03 | 0.281 | 0.044 |
| GEMS-MSD[3] | | | | | 0.007 | | | | | 0.071 | 0.429 |
| Less-severe | 56 (80%) | 568 (89%) | — | — | | 82 (80%) | 508 (86%) | — | — | | |
| Moderate-to-severe | 14 (20%) | 68 (11%) | 2.20 | 1.24, 3.89 | | 21 (20%) | 82 (14%) | 1.48 | 0.97, 2.26 | | |
| MVS+/- dysentery[4] | | | | | 0.722 | | | | | 0.644 | 0.992 |
| Mild with no dysentery | 46 (66%) | 400 (63%) | — | — | | 64 (62%) | 351 (59%) | — | — | | |
| Moderate/Severe or Dysentery | 24 (34%) | 236 (37%) | 0.91 | 0.55, 1.51 | | 39 (38%) | 239 (41%) | 0.92 | 0.64, 1.32 | | |
| Household sanitation | | | | | 0.497 | | | | | 0.005 | 0.046 |
| Improved | 55 (79%) | 482 (76%) | — | — | | 66 (64%) | 454 (77%) | — | — | | |
| Unimproved/Open defecation | 15 (21%) | 154 (24%) | 0.82 | 0.46, 1.46 | | 37 (36%) | 136 (23%) | 1.68 | 1.18, 2.41 | | |

*(Continued)*

**Table 3.** (Continued)

| Characteristic | Dry season | | | | | Rainy season | | | | | |
|---|---|---|---|---|---|---|---|---|---|---|---|
| | *Shigella +1* N = 70 | *Shigella -1* N = 636 | PR[2] | 95% CI[2] | p-value | *Shigella +1* N = 103 | *Shigella -1* N = 590 | PR[2] | 95% CI[2] | p-value | Effect Modifier p-value |
| Household hygiene | | | | | 0.823 | | | | | 0.645 | 0.790 |
| Wash hands without soap | 22 (31%) | 198 (31%) | — | — | | 31 (30%) | 165 (28%) | — | — | | |
| Wash hands with soap | 48 (69%) | 436 (69%) | 1.06 | 0.62, 1.81 | | 72 (70%) | 425 (72%) | 0.91 | 0.62, 1.34 | | |

[1] n (%); Median (Q1, Q3)

[2] PR = Prevalence Ratio, CI = Confidence Interval

[3] Defined as in Kotloff, Lancet GH, 2019. Moderate-to-severe diarrhea (MSD) defined as presenting to a health facility with diarrhea and severe or some dehydration (by WHO criteria), visible blood in stool, or inpatient admission.

[4] Defined as in Pavlinac, Vaccines, 2022 as a MVS of 9+ or presence of visible blood in stool.

This study also found that stunting was associated with *Shigella* in univariate analysis, although this did not persist in adjusted models. Other studies, including Lima et al. [26], have documented links between malnutrition and *Shigella*, likely due to compromised immunity and disrupted gut integrity in undernourished children [26]. Moreover, the bidirectional relationship between malnutrition and diarrheal diseases may perpetuate a cycle of vulnerability and illness [26].

Seasonal modification of risk factors for *Shigella*-attributable diarrhea emerged as a notable finding in our analysis. The association between *Shigella*-attributable diarrhea and wasting was stronger during the dry season but not seen in the rainy season. Children may be more likely to experience malnutrition during the dry season due to seasonal food insecurity, making them more susceptible to infections through compromised immune function. This is consistent with findings from a study in Haydom, Tanzania, which describe seasonal patterns of food insecurity, with the pre-harvest dry season significantly associated with lower birthweight, and an increased risk of hospitalization for early childhood acute malnutrition, [27]. The relatively lower circulation of other enteric pathogens in the dry season may make *Shigella* a more prominent cause of diarrhea, thereby strengthening its association with wasting [28].

Of note, the duration of diarrhea was marginally associated with *Shigella* during the dry season but not during the rainy season. Prior research has found that care-seeking for childhood diarrhoea in southern Malawi is influenced by cultural beliefs and access barriers [29], suggesting there may be a seasonal pattern to care-seeking that causes greater delays during the dry season.

Unimproved sanitation was associated with increased *Shigella* risk during the rainy season but not during the dry season. This is likely due to the increased movement of fecal contamination into living environments during the rainy season due to floods, consistent with previous studies in flood-prone areas such as Borbón, Ecuador, where poorly constructed open-pit latrines contributed to environmental contamination during heavy rains which exacerbates the spread of pathogens in areas with inadequate sanitation [30].

Vaccination emerged as a significant protective factor against *Shigella*-attributable diarrhea in the univariate model but not in multivariate models. Children who had received age-appropriate vaccinations were less likely to develop *Shigella*-related illness compared to children who did not receive age-appropriate vaccines or children whose vaccine status was unknown. This is consistent with previous studies that have identified vaccination status as a proxy indicator for broader health and socioeconomic advantages. For instance, research from Kenya and Tanzania indicated that children who were fully immunized were also more likely to live in households with improved sanitation, higher maternal education levels, and better access to healthcare services, all of which contribute to lower enteric disease risk [18,19]. This suggests that the protective association observed in the univariate model may not be solely due to the vaccines themselves, but also to the healthier living environments and practices of families who adhere to vaccination schedules [18,19]. While

children with unknown vaccination status may be a diverse group, they are are more likely to include those who are zero-dose or undervaccinated and, according to large multi-country analyses, are disproportionately from socioeconomically disadvantaged households with lower maternal education, less antenatal care, and reduced healthcare access. [31] These factors may increase susceptibility to infectious diseases including *Shigella*, explaining why we observed that children with unknown vaccine status have a higher risk of *Shigella* infection compared to children who received age-appropriate vaccines [31].

While culture identifies viable bacteria and is more specific to active infections, qPCR can detect *Shigella* DNA at lower levels, including when bacteria are non-viable. To improve the clinical relevance of qPCR-based detection, the EFGH study applied standardized laboratory methods [32]. Despite methodological differences, our findings showed consistent associations between most predictors and *Shigella* across both diagnostic methods. Notably, 43% of qPCR-positive cases were also culture-positive, supporting the clinical relevance of the qPCR-detected cases. qPCR is a more sensitive method, meaning that even though there is only 43% "overlap" with culture, it is identifying many *Shigella* infections that culture may miss. Additionally, qPCR enables detection of subclinical *Shigella* infections, which have been associated with significant linear growth faltering in young children [33]. As qPCR becomes more cost-effective, it may serve as a reliable alternative to culture in both research and surveillance settings [33].

This study had several strengths, including a large sample size, robust data collection and quality assurance procedures, and employed both culture and qPCR to detect *Shigella* infection. However, this study had limitations. There was a high proportion of children with missing vaccination data (32%). While missing vaccination data is common in pediatric research in low- and middle-income countries and is therefore a group of interest, by considering these children together we may be collapsing unobserved heterogeneity [34]. This was a facility-based study, and therefore we cannot draw conclusions about the predictors of diarrhea in the community overall, just among medically-attended diarrhea. In addition, we decided not to examine prior antibiotic use in this analysis due to the difficulty of interpreting model results based on high documented rates of AMR in *Shigella* and the potential unreliability of reported antibiotic type given by caregivers prior to enrollment [35]. Finally, the cross-sectional design of this study was able to capture associations between variables but cannot determine causality.

## Conclusion

These findings highlight the significant burden of *Shigella*-attributable diarrhea among young children aged 6–35 months in Malawi and emphasize the need for targeted interventions. Children who are stunted or wasted may be at higher risk for infection, underscoring the importance of addressing malnutrition as part of *Shigella* prevention strategies. Additionally, the risk of *Shigella* infections in children with moderate or severe wasting and unimproved sanitation may differ between the dry and rainy seasons, pointing to the need for seasonal interventions that focus on sanitation, water quality, and food security. The study also reinforces the importance of context-specific research and comprehensive public health efforts, including vaccination programs and improved sanitation, to reduce the prevalence of *Shigella* and improve health outcomes in vulnerable populations.

## Author contributions

**Conceptualization:** Vitumbiko Pablo Yagontha Munthali, Donnie Mategula.

**Data curation:** Vitumbiko Pablo Yagontha Munthali, Olivia Schultes.

**Formal analysis:** Vitumbiko Pablo Yagontha Munthali, Olivia Schultes, Donnie Mategula.

**Funding acquisition:** Vitumbiko Pablo Yagontha Munthali, Khuzwayo C Jere, Jennifer Cornick, Nigel A Cunliffe, Agra Thindwa, Landwel Mwale, Patricia B Pavlinac.

**Investigation:** Vitumbiko Pablo Yagontha Munthali, Olivia Schultes, Desiree Witte, Khuzwayo C Jere, Jennifer Cornick, Nigel A Cunliffe, Clement Lefu, Maureen Ndalama, Landwel Mwale, Stephen Munga, Donnie Mategula.

**Methodology:** Vitumbiko Pablo Yagontha Munthali, Olivia Schultes, Stephen Munga, Donnie Mategula.

**Project administration:** Vitumbiko Pablo Yagontha Munthali, Olivia Schultes, Desiree Witte, Khuzwayo C Jere, Nigel A Cunliffe, Clement Lefu, Maureen Ndalama, Agra Thindwa, Landwel Mwale, Stephen Munga, Donnie Mategula.

**Resources:** Vitumbiko Pablo Yagontha Munthali, Khuzwayo C Jere, Jennifer Cornick, Nigel A Cunliffe, Maureen Ndalama, Agra Thindwa.

**Software:** Vitumbiko Pablo Yagontha Munthali, Olivia Schultes.

**Supervision:** Vitumbiko Pablo Yagontha Munthali, Olivia Schultes, Desiree Witte, Khuzwayo C Jere, Jennifer Cornick, Nigel A Cunliffe, Clement Lefu, Maureen Ndalama, Agra Thindwa, Landwel Mwale, Stephen Munga, Donnie Mategula.

**Validation:** Vitumbiko Pablo Yagontha Munthali, Olivia Schultes, Latif Ndeketa, Stephen Munga, Donnie Mategula.

**Visualization:** Vitumbiko Pablo Yagontha Munthali, Latif Ndeketa, Stephen Munga, Donnie Mategula.

**Writing – original draft:** Vitumbiko Pablo Yagontha Munthali, Olivia Schultes, Desiree Witte, Khuzwayo C Jere, Jennifer Cornick, Nigel A Cunliffe, Clement Lefu, Maureen Ndalama, Landwel Mwale, Latif Ndeketa, Stephen Munga, Donnie Mategula.

**Writing – review & editing:** Vitumbiko Pablo Yagontha Munthali, Olivia Schultes, Desiree Witte, Khuzwayo C Jere, Jennifer Cornick, Nigel A Cunliffe, Clement Lefu, Maureen Ndalama, Landwel Mwale, Latif Ndeketa, Stephen Munga, Donnie Mategula.

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
