## [Decision Letter · Decision Letter 0]

6 Jan 2026

PGPH-D-25-03065

Risk factors associated with *Shigella*  diarrhea in 6-35-month-olds: A Cross-Sectional Study, Malawi, 2022-2024. diarrhea in 6-35-month-olds: A Cross-Sectional Study, Malawi, 2022-2024. diarrhea in 6-35-month-olds: A Cross-Sectional Study, Malawi, 2022-2024. diarrhea in 6-35-month-olds: A Cross-Sectional Study, Malawi, 2022-2024.

Dear Dr. Yagontha Munthali,

Thank you for submitting your manuscript to PLOS Global Public Health. After careful consideration, we feel that it has merit but does not fully meet PLOS Global Public Health’s publication criteria as it currently stands. Therefore, we invite you to submit a revised version of the manuscript that addresses the points raised during the review process.

Two reviewers have evaluated your manuscript and provided their comments below. In particular, please provide more detail in the methods section on the microbiological methods for Shigella detection and add information on any missing critical variables e.g. in the table footnotes. In addition and given the high missingness for vaccination status, conducting a sensitivity analysis for the multivariable models while excluding vaccination would aid with the interpretation of the study findings.

We look forward to receiving your revised manuscript.

Kind regards,

Ioana D. Olaru

Academic Editor

Journal Requirements:

1. Please provide a detailed online Financial Disclosure statement. This is published with the article. It must therefore be completed in full sentences and contain the exact wording you wish to be published.

a) State the initials, alongside each funding source, of each author to receive each grant. For example: “This work was supported by the National Institutes of Health (####### to AM; ###### to CJ) and the National Science Foundation (###### to AM).”

For more information, please go to our submission guidelines:

https://journals.plos.org/globalpublichealth/s/submission-guidelines#loc-financial-disclosure-statement

2. Please ensure that the funders and grant numbers match between the Financial Disclosure field and the Funding Information tab in your submission form. Note that the funders must be provided in the same order in both places as well.

3. Please update your online Competing Interests statement. If you have no competing interests to declare, please state: “The authors have declared that no competing interests exist.”

5. Please ensure that the Title in your manuscript and the Title in your online submission form are the same.

6. Please include a separate legend or caption for Figure 1 in your manuscript.

Additional Editor Comments (if provided):

Reviewers' comments:

Reviewer's Responses to Questions

**Comments to the Author**

1. Does this manuscript meet PLOS Global Public Health’s publication criteria? Is the manuscript technically sound, and do the data support the conclusions? The manuscript must describe methodologically and ethically rigorous research with conclusions that are appropriately drawn based on the data presented.? Is the manuscript technically sound, and do the data support the conclusions? The manuscript must describe methodologically and ethically rigorous research with conclusions that are appropriately drawn based on the data presented.

Reviewer #1: Yes

Reviewer #2: Yes

2. Has the statistical analysis been performed appropriately and rigorously?

Reviewer #1: Yes

Reviewer #2: Yes

3. Have the authors made all data underlying the findings in their manuscript fully available (please refer to the Data Availability Statement at the start of the manuscript PDF file)?

The PLOS Data policy requires authors to make all data underlying the findings described in their manuscript fully available without restriction, with rare exception. The data should be provided as part of the manuscript or its supporting information, or deposited to a public repository. For example, in addition to summary statistics, the data points behind means, medians and variance measures should be available. If there are restrictions on publicly sharing data—e.g. participant privacy or use of data from a third party—those must be specified.requires authors to make all data underlying the findings described in their manuscript fully available without restriction, with rare exception. The data should be provided as part of the manuscript or its supporting information, or deposited to a public repository. For example, in addition to summary statistics, the data points behind means, medians and variance measures should be available. If there are restrictions on publicly sharing data—e.g. participant privacy or use of data from a third party—those must be specified.

Reviewer #1: No

Reviewer #2: Yes

4. Is the manuscript presented in an intelligible fashion and written in standard English?

Reviewer #1: Yes

Reviewer #2: Yes

Reviewer #1: This paper examines factors associated with Shigella-attributed diarrhea among children aged 6–35 months in Malawi, including a novel assessment of seasonal effect modification. The analyses are technically rigorous and appropriately applied to the observational dataset, and the findings provide valuable evidence to guide targeted interventions, including the forthcoming Shigella vaccine rollout. I recommend publication pending a few minor revisions noted below.

• The introduction explicitly situates the burden of Shigella in an economic context, but are there any other lenses, perhaps more human-centric, through which we can think about the implications of this burden?

• Please include references for the categorization methods of diarrhea severity and WASH in the “Predictor Variables” section of the Methods.

• How did you approach producing age group bins for this study? Was it data driven or decided a priori based on some contextual motivation?

• Please add a statement justifying Poisson regression as the chosen analytic method.

• Given that some samples were tested by culture, some by qPCR, and some by both it would be beneficial to add more clarification in the methods about the different testing procedures and results classification. For the samples tested by both methods, what happened if one result was positive and the other negative? In the discussion you state “Notably, 43% of qPCR-positive cases were also culture-positive, supporting the clinical relevance of the qPCR-detected cases”, however only 43% overlap between the two methods actually seems quite low – can you point to any other studies that have looked at this?

• I believe when using generalized estimating equations (GEE) to account for clustering it is standard to report the number of clusters and distribution of cluster size.

• These analyses rely on an assumption of missing data at random/completely at random, however the complete case analysis conducted excludes 32% of children with missing vaccination data. Please elaborate on this missingness in the limitations beyond reduction in sample size to include the potential bias introduced if the data is not in fact missing randomly. Alternatively, it may be worthwhile to consider inclusion of the observations with unknown vaccination status as a third category – which may still have relevant interpretation given the reality of often not knowing children’s vaccination status when designing interventions.

• Was there any consideration of prior antibiotic use among patients reporting to the clinic with diarrhea? Please elaborate how this may or may not influence these results (perhaps in the limitations).

• Please standardize spelling of “enrollment” throughout the manuscript (sometimes one vs two ls).

• In Table 1 the Wasting “None” group percentage needs a decimal instead of a comma.

Reviewer #2: The findings are interesting but it need through revision considering the following critical points.

• Clearly mention the inclusion and exclusion criteria for selection of patients in the current study?

• What was the limitation of the study?

• Shigella were isolated and identified using culturing. What was the specie distribution of Shigella?

• Mention the duration of the study (months/years) in the abstract.

• Briefly describe how the culture and qPCR were used for detection of Shigella. Is Shigella DNA directly detected in fecal sample or it is detected from culture?

• Define the criteria of Household drinking water source categorization: Improved, Unimproved??

• The abbreviations used in the tables should be defined in the table’s foot note.

**Do you want your identity to be public for this peer review?** For information about this choice, including consent withdrawal, please see our Privacy Policy..

Reviewer #1: No

Reviewer #2: No

---

## [Editor Report · Decision Letter 1]

24 Feb 2026

Risk factors associated with *Shigella* diarrhea in 6-35-month-olds: A Cross-Sectional Study, Malawi, 2022-2024.diarrhea in 6-35-month-olds: A Cross-Sectional Study, Malawi, 2022-2024.diarrhea in 6-35-month-olds: A Cross-Sectional Study, Malawi, 2022-2024.diarrhea in 6-35-month-olds: A Cross-Sectional Study, Malawi, 2022-2024.

PGPH-D-25-03065R1

Dear Mr. Yagontha Munthali,

We are pleased to inform you that your manuscript 'Risk factors associated with *Shigella* diarrhea in 6-35-month-olds: A Cross-Sectional Study, Malawi, 2022-2024.' has been provisionally accepted for publication in PLOS Global Public Health.diarrhea in 6-35-month-olds: A Cross-Sectional Study, Malawi, 2022-2024.' has been provisionally accepted for publication in PLOS Global Public Health.diarrhea in 6-35-month-olds: A Cross-Sectional Study, Malawi, 2022-2024.' has been provisionally accepted for publication in PLOS Global Public Health.diarrhea in 6-35-month-olds: A Cross-Sectional Study, Malawi, 2022-2024.' has been provisionally accepted for publication in PLOS Global Public Health.

Best regards,

Ioana D. Olaru

Academic Editor